# Uncovering the hidden yeast diversity in fermented coffee: Insights from a shotgun metagenomic approach

Katherine Bedoya-Urrego[1☉], Aida Esther Peñuela-Martínez[2☉], Juan F. Alzate [1,3]*

**1** Centro Nacional de Secuenciación Genómica CNSG, Sede de Investigación Universitaria-SIU, Universidad de Antioquia, Medellín, Colombia, **2** National Coffee Research Center, Cenicafé, Manizales, Colombia, **3** Departamento de Microbiología y Parasitología, Facultad de Medicina, Universidad de Antioquia, Medellín, Colombia

☉ These authors contributed equally to this work.
* jfernando.alzate@udea.edu.co

## Abstract

Yeasts play a pivotal role in coffee fermentation, shaping microbial succession and contributing to the development of final flavor profiles. Despite their importance, yeast taxonomy in this context remains poorly resolved. Traditional classification methods often result in misidentifications due to the limited resolution of classical microbiological techniques and the rapidly evolving taxonomic framework driven by advances in phylogenomic. Moreover, the diversity of budding yeasts in coffee fermentations remains underexplored using high-resolution approaches such as metagenomics. To address this gap, we applied a shotgun metagenomic strategy and reconstructed metagenome-assembled genomes (MAGs) from multiple coffee fermentation samples and, using a robust phylogenomic framework based on 832 conserved single-copy genes. We confidently classified 22 yeast MAGs within the subphylum Saccharomycotina. These included well-known taxa such as *Pichia kluyveri*, *Hanseniaspora* spp., *Torulaspora delbrueckii*, and members of the *Kurtzmaniella* clade. Most MAGs were placed in strongly supported monophyletic groups (ultrafast bootstrap = 100), with short intra-clade branch lengths indicative of intraspecific variation. *Pichia kluyveri* emerged as the most abundant and widespread species, detected in all analyzed metagenomes, followed by *Hanseniaspora* spp. Our results underscore the power of high-resolution phylogenomic for classifying yeast MAGs and highlight the ecological importance of Pichia, *Hanseniaspora*, *Torulaspora*, and *Kurtzmaniella* in spontaneous coffee fermentations.

## Introduction

Coffee is one of the most widely consumed beverages globally and holds significant economic importance [1,2]. Currently, coffee is produced in more than 60 countries,

**Data availability statement:** Raw Reads in teh NCBi SRA database https://www.ncbi.nlm.nih.gov/sra/PRJNA1305218. The sample identifiers are SRR34971509, SRR34971508, SRR34971516, SRR34971515, SRR34971514, SRR34971513, SRR34971512, SRR34971511, and SRR34971510.

**Funding:** This research was developed under the project: Experimental Development for the Competitiveness of the Coffee Sector of the Department of Quindío, code 2017000100099, financed by the General System of Royalties, Gobernación de Quindío in agreements signed with the National Colombian Coffee Growers Federation (Cenicafé—Crossref Funder ID 100019597). No. 002-of-2020. The funders had no role in study design, data collection and analysis, decision to publish, or preparation of the manuscript.

**Competing interests:** The authors have declared that no competing interests exist.

with Brazil being the world's leading producer, followed by Vietnam and Colombia [1]. Collectively, these three countries contribute over 54.5% of global coffee production, underscoring their central role in the international coffee supply chain [3,4]. The production process requires microbial fermentation to facilitate the elimination of the residual ƒmucilaginous layer that surrounds the coffee beans [4]. The fermentation process is carried out by the indigenous microbiota including yeasts, bacteria, and filamentous fungi, producing a wide range of metabolites like esters, ketones, alcohols, acids, and aldehydes that have a substantial impact on coffee sensory qualities [3,5]. This microbiological process occurs spontaneously and is driven primarily by bacterial and fungal communities, which significantly influence the sensory quality of coffee. Its dynamics depend on several factors, including time, temperature, and water exchange [6,7].

Yeasts are detected in all types of coffee fermentations revealing pectinolytic activity and secondary metabolites production that affect the coffee quality [8]. The yeast genera reported as dominant in coffee fermentation in different locations are *Hanseniaspora*, *Candida*, *Kluyveromyces*, *Pichia*, *Saccharomyces*, *Rhodotorula*, *Debaryomyces,* and *Schizosaccharomyces [*9,10*]*. Coffee fermentation samples are particularly complex due to the environments in which they are found, and the diversity of substrates and the production of microbial metabolites can lead beneficial or undesirable effects [11]. Accordingly, a deeper exploration is necessary to determine the specific genera and species related to coffee fermentation. Additionally, understanding the functions of these yeasts, their special characteristics, and their relationship with coffee compounds is crucial for advancing knowledge of yeasts and their biotechnological potential in coffee fermentation [9].

Taxonomic assignment in yeasts remains challenging, with frequent reports of misidentified species and genera [12]. Metagenomics is a powerful approach for exploring yeast diversity in food fermentation processes, providing an integrated and less biased view of this taxonomic group and their functional ecology [13,14]. Previous metagenomic studies have revealed dynamic changes in gene repertoires and temporal shifts in both bacterial and yeast communities during coffee fermentation. These variations, influenced by geographical origin and fermentation type, have been shown to markedly affect the metabolic potential of the microbiota and, consequently, the sensory quality of coffee [15]. Metagenomic binning enabled the reconstruction of metagenome-assembled genomes (MAGs), allowing a more precise assessment of yeast contributions to the fermentation process and providing high-resolution taxonomic classification [16,17].

In this study, we applied a shotgun metagenomic approach to characterize yeast diversity and reconstruct MAGs from multiple industrial-scale coffee fermentation batches in Colombia. our aim was to recover and perform phylogenomic analyses to identify, isolate, and accurately classify yeast communities present in industrial-scale fermentation batches of *Coffea arabica* from Colombia's central coffee zone. We developed a robust phylogenomic framework based on 832 conserved single-copy protein-coding genes spanning the Saccharomycotina subphylum. This strategy enabled the confident characterization of yeast communities and provided strong evidence for the identity and prevalence of key taxa driving coffee fermentation.

## Materials and methods

### Coffee fermentation samples and collection permission

The fermented coffee samples (*Coffea arabica* L. Castillo®) produced in seven farms from the Department of Quindío, Colombia were collected during the harvest season of the second semester of 2021. The samples were derived from traditional fermentations carried out on each farm, typically performed overnight, with variations in the process time and use of water. The S1 Table describes the fermentation type, the location, and mean annual average environmental conditions of each farm.

Sample collection was conducted under the valid permit issued by Resolution No. 1508 of September 6, 2018. The permit was granted by the NATIONAL ENVIRONMENTAL LICENSING AUTHORITY OF COLOMBIA (ANLA) to the Post-Harvest Coffee Research Group (COL0006634) in the implementation of the project POS101031, "Characterization of the microbial composition in the fermentation of coffee produced in the department of Quindío"; This was within the broader framework of the project "Experimental Development for the Competitiveness of the Coffee Sector of the Department of Quindío"; code 2017000100099, in agreements signed with the National Colombian Coffee Growers Federation (Cenicafé).

### DNA extraction and sequencing

Fermented samples were collected at the end of the fermentation process and stored at −80°C until further processing. Genomic DNA (gDNA) was extracted using the DNeasy PowerLyzer PowerSoil Kit (QIAGEN, Hilden, Germany) following the manufacturer's instructions. DNA concentration and purity were assessed using a NanoDrop™ 2000 spectrophotometer (Thermo Scientific™) by measuring absorbance at 260 nm and 280 nm. DNA integrity was evaluated by electrophoresis on a 1% agarose gel. High-quality gDNA samples were stored at −20°C prior to metagenomic sequencing. Shotgun metagenomic sequencing was performed at Macrogen, Inc. (Seoul, South Korea) using the Illumina NovaSeq 6000 platform with 150 bp paired end reads. Shotgun libraries were prepared with the TruSeq Nano DNA Library Preparation Kit (Illumina).

### Read quality trimming and assembly

The adapters, singletons, and poor-quality reads (<Q30) were filtered using CUTADAPT software with the following parameters: -j 50 -q 30 -m 70 --max-n 0 (v 3.5) [18]. MetaSPADES (version 3.14.1) [19] was employed for shotgun metagenome assembly with specified flags -t 40 -m 160, testing k-mer lengths (-k) of 55, 77, and 99 bases.

### Metagenome taxonomic analysis

Metagenomic taxonomic assignment was performed using MEGAN (v.6.25) [20] to characterize the microbial composition of the coffee fermentation samples. Raw paired-end reads were first quality trimmed with *cutadapt* to remove adapters and low-quality bases. Subsequently, forward and reverse reads were merged into longer single-end reads using FLASH tool (v1.2.11) [21], thereby improving taxonomic resolution by providing longer sequence context. The merged high-quality reads were then aligned with DIAMOND (v2.0.14.152) tool and DAA files were imported into MEGAN for taxonomic assignment based on the lowest common ancestor (LCA) algorithm (default parameters: Min Score: 50, Top Percent: 10%, and Max Expected: 0,001), allowing reliable classification of reads across taxonomic levels. Taxonomic profiles generated by MEGAN were used for downstream statistical analyses and visualization in R.

### Binning, MAGs isolation, and phylogenomic analysis

Metagenome-assembled genomes (MAGs) were recovered through binning of contigs using MetaBAT2 (v2.12.1) [22]. Sequencing depth for each MAG was estimated by mapping quality-filtered reads to the corresponding binned scaffolds

generated by metaSPAdes, using Bowtie2 (v2.3.4.3) with default alignment parameters [23]. The resulting alignments were processed into sorted and indexed BAM files using SAMtools [24].

Bins corresponding to yeasts were identified through sequence alignment against a curated yeast database using the BLASTN tool [25], applying stringent filters (E-value = 0 and bit score > 10,000). MAG quality was assessed using BUSCO [26] and EUKCC software (v2.1.3) [27] with default parameters. Sequencing depth and breadth of MAG coverage were calculated by mapping quality-filtered reads to the MAGs using Bowtie2 [23], followed by coverage estimation with SAMtools [24] (S2 Table). For phylogenomic reconstruction, reference yeast genomes available in the RefSeq database of GenBank were selected. See S3 Table for genome accession numbers. Single-copy orthologous proteins were identified across these genomes using SONICPARANOID software [28], resulting in the detection of 871 conserved proteins shared among at least 50% of the included yeast genomes. A concatenated super matrix was then constructed by aligning each orthologous protein loci using the MAFFT algorithm [29]. The program *catsequences* was used to concatenate individual protein alignments (https://github.com/ChrisCreevey/catsequences). In cases where certain protein loci were absent in specific genomes, alignment gaps were inserted in the corresponding regions to maintain matrix structure.

The protein super matrix was used to infer a maximum likelihood phylogenetic tree with IQ-TREE2, employing 5,000 ultrafast bootstrap replicates [30]. Protein model applied to the super matrix was LG4M [31]. *Schizosaccharomyces pombe* was designated as the outgroup to root the tree. The resulting phylogeny was visualized and annotated using the FigTree software (v1.4.4).

### Functional prediction

Gene prediction was carried out with AUGUSTUS software (v 3.4.0) [32] to identify coding sequences and extract the corresponding protein sequences from each yeast MAG. For each assembly, predictions were generated using the preconfigured models *Saccharomyces cerevisiae* S288C, *S. cerevisiae* RM11-1a, and *Kluyveromyces lactis*, with the following parameters: --softmasking=off, --UTR=off, and --noInFrameStop = true. The resulting GFF3 files for each MAG were then merged using gffcompare tool [33] to consolidate gene models and ensure consistent gene structure annotation across datasets.

These predicted proteins were then subjected to functional annotation using eggNOG-mapper, enabling assignment of orthologous groups, KEGG Orthology (KO) identifiers, and functional categories. KEGG pathway reconstruction was performed to identify metabolic pathways relevant to coffee fermentation, with particular emphasis on those associated with the pectin degradation, production of volatile compounds, and key organic acids.

### Statistical and graphical analyses

Statistical and graphical analyses were conducted in the R environment (v.4.x; R Core Team, 2024). Data wrangling and summarization were performed using the *tidyverse* framework, primarily with the *dplyr* and *tidyr* packages, to reshape, filter, and aggregate metagenomic data. Visualization of results was carried out with *ggplot2*, which was used to generate barplots, boxplots, scatterplots, and phylogenomic placement summaries with customized aesthetics for clarity. The *ggpubr* package was employed for comparative statistical testing (e.g., Wilcoxon rank-sum tests) and figure arrangement, while *scales* and *RColorBrewer* were used for color palettes and axis formatting. All plots were produced in vector format to ensure high-quality resolution for publication.

## Results

### Physicochemical characteristics of coffee fermentation samples

In this study, we analyzed coffee fermentation samples produced using different fermentation strategies, durations, and environmental conditions in a coffee-growing region of Colombia. Traditional fermentation was the most common approach, typically lasting between 15 and 19 hours, and was associated with lower pH values (3.29–3.54), suggesting

a more acidic environment potentially favorable to acid-tolerant microbial taxa. Notably, sample CF13 deviated from this trend, displaying a markedly higher pH of 4.50. The temperatures in the traditional method were stable, approximately 20–22°C. In contrast prolonged fermentation (sample CF08) and underwater fermentation (CF02 and CF12) were associated with longer durations (≥39 h) and characterized by higher pH values (up to 4.11 in CF12) and elevated fermentation temperatures (e.g., 26.3°C in CF08). The region in which sample were collected has an altitude ranged from 1,288 m to 1,832 m above sea level. While environmental temperature and relative humidity remained fairly consistent across sites (18–21°C and 77–83%, respectively).

## Metagenome sequencing, QC, and assembly metrics

The raw reads from the nine metagenomes totaled 44.1 GB, representing an average of 61,152,198 paired-end reads per sample. After quality trimming and filtering, over 92% of the raw bases passed the Q30 quality threshold and the minimum length filter of 70 bases. A total of nine metagenomes were assembled, one per fermentation sample, with a median assembly size of 603 Mb, ranging from 316 Mb to 836 Mb (S1 Fig, panel A). The number of scaffolds per assembly had a median of 1,298,731, with values between 675,180 and 1,943,561 (S1 Fig, panel B). Scaffold continuity, measured by N50, had a median of 604 bp and ranged from 511 bp to 766 bp, reflecting the expected fragmentation typical of complex environmental metagenomes (S1 Fig, panel C). The largest scaffold in each assembly ranged from 268 kb to 705 kb, with a median length of 372 kb. GC content was relatively consistent across samples, with a median of 44.9% and a range from 41.3% to 51.6%. Sample CF14 displayed the highest GC content (51.6%), standing out as an outlier (S1 Fig, panel D). The proportion of ambiguous bases was low across all samples, ranging from 0.06% to 0.13% (S1 Fig, panel E). Despite the high number of assembled scaffolds, only a small fraction exceeded 1 kb in length, ranging from 4.4% to 7.8% across all samples. The longest and most abundant scaffolds likely correspond to chromosomal fragments from dominant organisms within the microbial communities. Plotting scaffold length against sequencing depth provides a visual representation of scaffold clusters, which may reflect the genomic architecture of the most abundant taxa and guide efforts to recover metagenome-assembled genomes (MAGs), particularly from prevalent yeast populations. Clusters of scaffolds potentially originating from the same genome appear as groups aligned along the x-axis, sharing similar sequencing depth and GC content profiles, suggesting consistent similar genomic origin and coverage characteristics (S2 Fig).

## Metagenome binning

Metagenomic binning was performed on all samples using the software METABAT2. The number of recovered bins (BINs) per sample varied between 38 and 70 across the coffee fermentation metagenomes. The median assembly size of the BINs per sample ranged from 409 kb to 12.7 Mb. While the smallest BINs in all samples were approximately 200 kb, the largest BINs generally ranged from 9 Mb to 13 Mb. Notably, sample CF13 produced an exceptionally large BIN of 72 Mb. The median number of scaffolds per BIN ranged from 36 to 139, with individual BINs containing as few as 1 scaffold and up to a maximum of 17,212 scaffolds. The N50 values—a measure of assembly continuity—had median values ranging from 9,866 bp to 26,320 bp across samples. The best N50 observed per BIN ranged from 143 kb to 466 kb. GC content analysis further illustrated the taxonomic and genomic diversity of the recovered bins. Median GC values across samples ranged from 40.5% to 51.4%, with individual BINs spanning a wide range from 22.4% to 73.6%, reflecting the heterogeneity of the underlying microbial communities (S3 Fig).

## Taxonomical assignment of MAGs

To identify metagenome-assembled genomes (MAGs) corresponding to putative yeasts and to generate a preliminary taxonomic profile of the yeast community present in the fermented coffee samples, we initially classified *de novo* assembled scaffolds using MEGAN (MEtaGenome ANalyzer). As expected, MEGAN results indicated that most assembled scaffolds originated from bacteria and Coffea plant sequences, followed by yeasts. Notably, a substantial proportion of scaffolds

were also assigned to filamentous fungi, particularly *Fusarium*. Among the most prevalent yeast genera identified by MEGAN package were *Hanseniaspora, Wickerhamomyces, Meyerozyma, Torulaspora, Pichia, Wickerhamiella, Candida-Lodderomyces clade*, and *Debaryomyces*, among others (Fig 1). However, as MEGAN relies on alignments to public databases that may contain misidentified sequences, we employed a more robust and accurate taxonomic strategy.

To refine the identification of yeast MAGs, we performed a BLASTN search against a custom database of reference genomes from budding yeasts (Saccharomycotina) using the bins recovered with METABAT2. Bins with significant hits

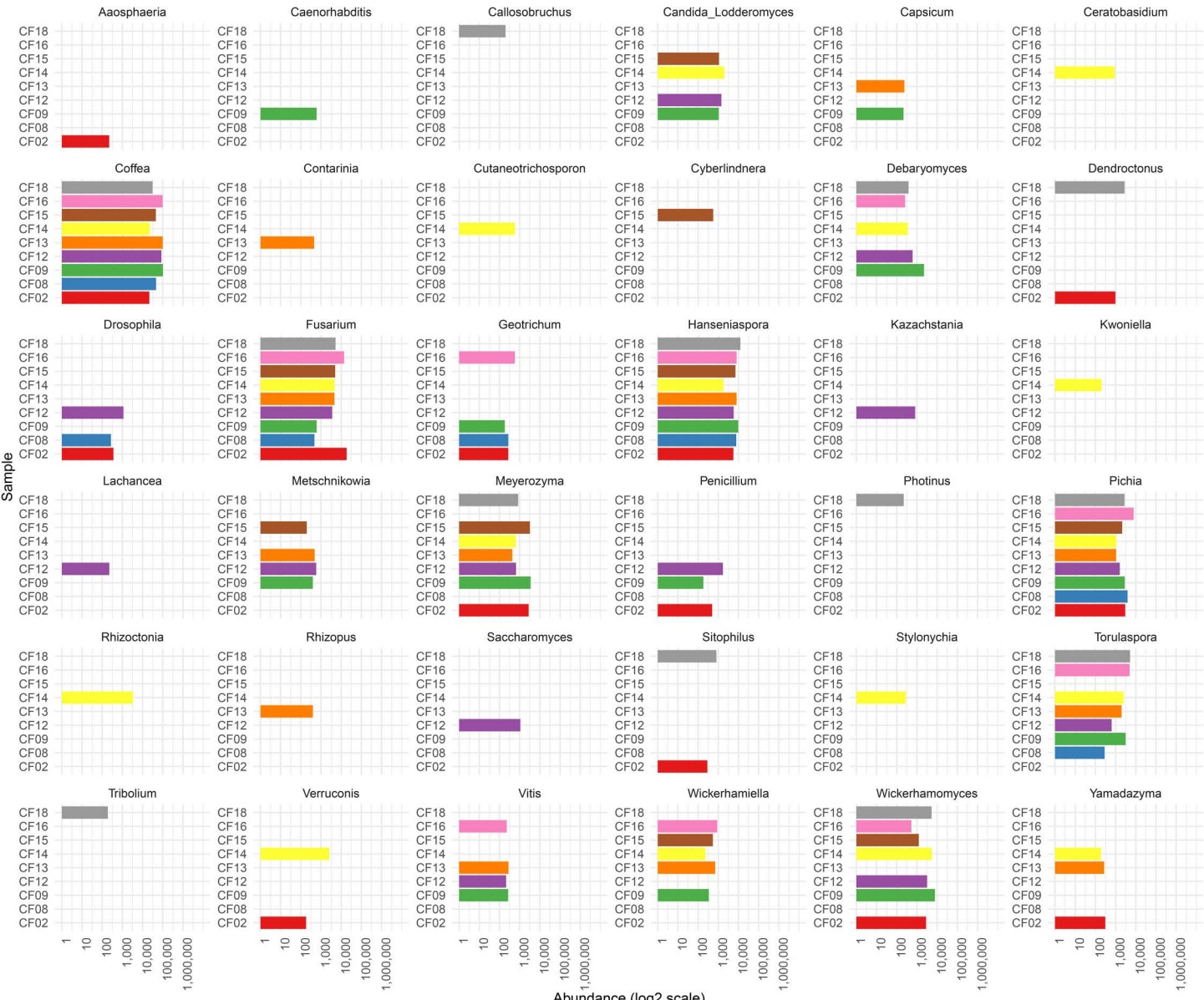

**Fig 1. Taxonomic profile of Eukaryote genera identified by MEGAN metagenome analyzer across coffee fermentation samples.** The bar plot in the x-axis shows the relative abundance (log₂-transformed) of each genus detected in metagenomic datasets, based on MEGAN metagenome analyzer taxonomic assignments. Each facet represents a different genus (Taxon), with bar heights corresponding to the abundance (number of scaffolds per taxa) per sample. Colors correspond to individual fermentation samples (y-axis).

exceeding defined thresholds were selected and further evaluated via phylogenomic analysis to confirm their taxonomic placement. As a measure to reduce noise in the phylogenomic reconstruction, we excluded bins whose genome size exceeded the median size of the assigned genus by more than 10%, based on reference genomes from the NCBI Datasets database. After filtering, a total of 22 high-confidence yeast MAGs were retained for downstream analysis (S3 Table).

To further assess the quality of the yeast MAGs, we mapped the cleaned reads back to the assembled genomes and quantified sequencing depth using the per-base depth functionality in SAMtools. As an additional measure of genome consistency, we also calculated the number of alternate alleles detected in each MAG, which serves as an indicator of intraspecies genetic heterogeneity. The number of alternate alleles varied considerably, ranging from 891 in *Kurtzmaniella* sp. to 177,693 in *Pichia kluyveri*, suggesting the coexistence of multiple strains of the same species within the same fermentation batch—particularly evident in *Pichia kluyveri*. As expected, per-base sequencing depth estimates were higher than those reported by MetaSPAdes, which bases its coverage estimates on k-mer frequencies. Coverage values across MAGs ranged from 6× to 142× (S3 Table).

To ensure precise taxonomic resolution, we constructed a phylogenomic framework using a broad set of Saccharomycotina reference genomes. BUSCO was used to identify conserved single-copy orthologs, and SONICPARANOID was applied to further refine the dataset by excluding duplicated or paralogous proteins. To account for protein loci variability, only orthologous present in at least 50% of the analyzed strains were included, resulting in a robust matrix of 832 protein-coding genes for phylogenetic reconstruction.

The resulting topology recapitulated the most recent class-level organization of the subphylum Saccharomycotina, including the classes *Lipomycetales*, *Trigonopsidales*, *Dipodascales*, *Saccharomycodales*, *Saccharomycetales*, *Phaffomycetales*, *Ascoideales*, *Pichiales*, and *Serinales* [34]. All class-level nodes were strongly supported with values of UFB support = 100, except for the *Saccharomycetes* clade (UFB support = 97). At the family level, most clades were monophyletic and supported by UFB support = 100. A notable exception was *Debaryomycetaceae*, which appeared split into two clades interrupted by *Metschnikowiaceae*. Although the combined clade received UFB support = 100, the internal node separating *Debaryomycetaceae* had low support (UFB support = 61), suggesting unresolved lineage boundaries. Similarly, *Dipodascaceae* appeared nested within *Trichomonascaceae*, with high confidence (Fig 2).

Despite these topological conflicts, well-recognized genera such as *Saccharomyces*, *Pichia*, *Wickerhamomyces*, *Debaryomyces*, *Hanseniaspora*, *Torulaspora*, and *Meyerozyma* were consistently recovered as monophyletic and placed within their expected families (UFB support = 100), reinforcing the robustness of our phylogenomic framework for accurate yeast MAG classification (Fig 2). The complete, uncollapsed tree is presented in Supplementary S4 Fig.

## Species-level assignment of yeast MAGs

As a quality control measure, one MAG exhibiting suspiciously long branch in the phylogenomic tree was removed to avoid including chimeric or misassembled genomes that could introduce noise into the phylogeny.

The phylogenomic analysis enabled confident species-level assignment for most of MAGs. Most belonged to well-established genera and were placed with high bootstrap support (UFB support = 100). Within Pichiaceae, nine MAGs were assigned to *Pichia kluyveri*, clustering tightly with the reference genome *P. kluyveri* (GCA_030062975) in a well-supported clade (UFB support = 100). Intra-clade distances ranged from 0.0008 to 0.0098, with a short distance to the reference genome (0.0070), indicating high similarity. In contrast, the closest sister species *P. heedii* was separated by a distance of 0.1464, and other species such as *P. kudriavzevii* were even more distant (e.g., 0.2906), confirming the conspecific status of the MAGs (Fig 3).

Eight MAGs were assigned to the class Saccharomycetes, six to *Saccharomycodaceae* and two to *Saccharomycetaceae*. All six *Saccharomycodaceae* MAGs clustered within the genus *Hanseniaspora*. Three MAGs grouped with *H. opuntiae*, forming a clade with UFB = 100. They displayed minimal divergence (0.0011–0.016 substitutions per site), confirming

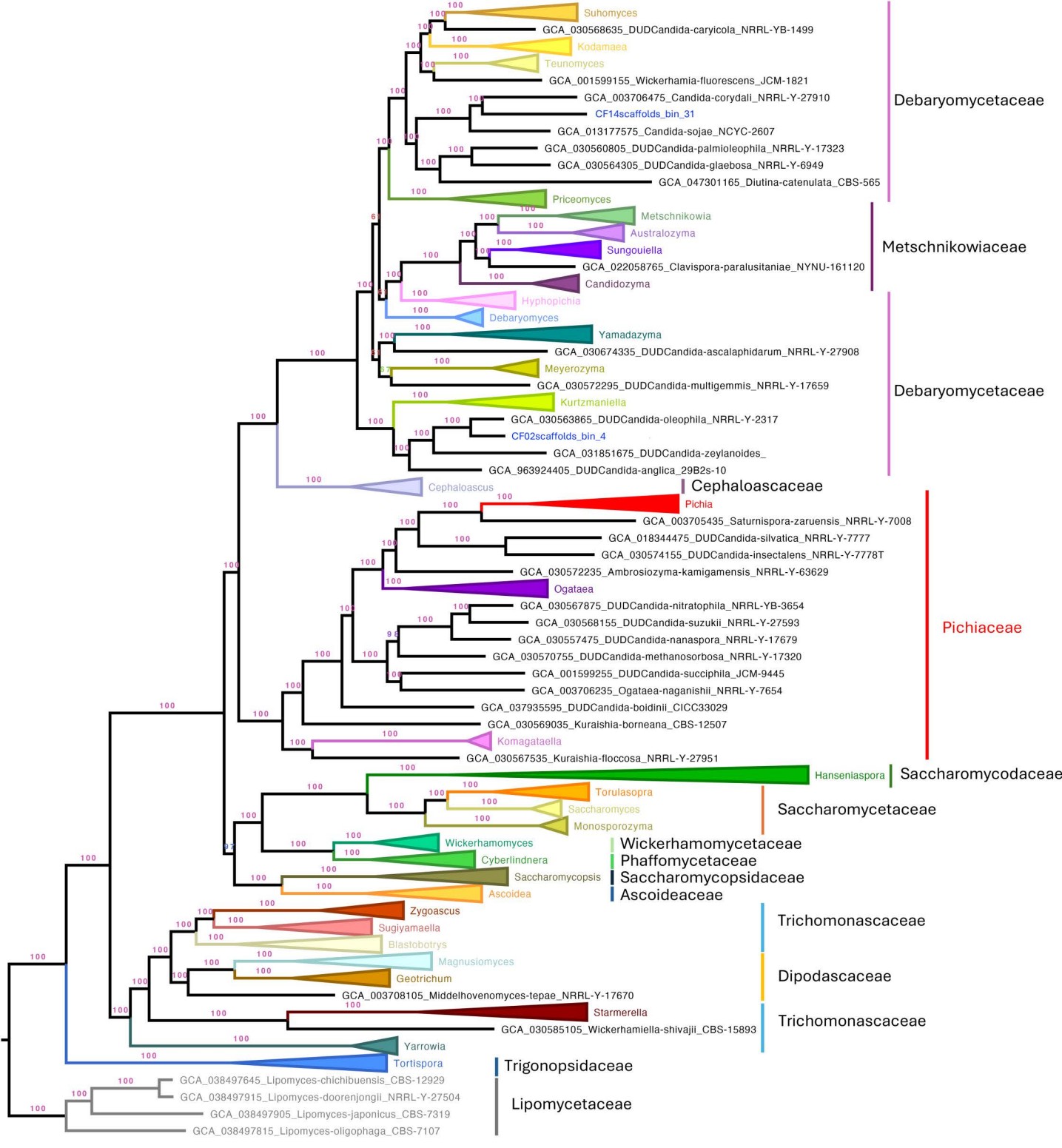

**Fig 2. Maximum-likelihood phylogenomic tree of Saccharomycotina constructed from 832 conserved single-copy orthologous proteins across representative genomes.** Protein loci were selected based on their presence in at least 50% of the analyzed taxa, and paralogs were excluded to ensure accurate gene representation. The tree recapitulates the latest class-level taxonomy within Saccharomycotina, including nine major classes

such as Lipomycetales, Trigonopsidales, and Saccharomycetales. Ultrafast bootstrap (UFB) support values are shown in respective branches, with most class and family clades receiving full support (UFB support = 100). Notable exceptions include the Saccharomycetes clade (UFB support = 97), a split within Debaryomycetaceae (internal node UFB support = 61). Vertical bars indicate the taxonomic boundaries of families.

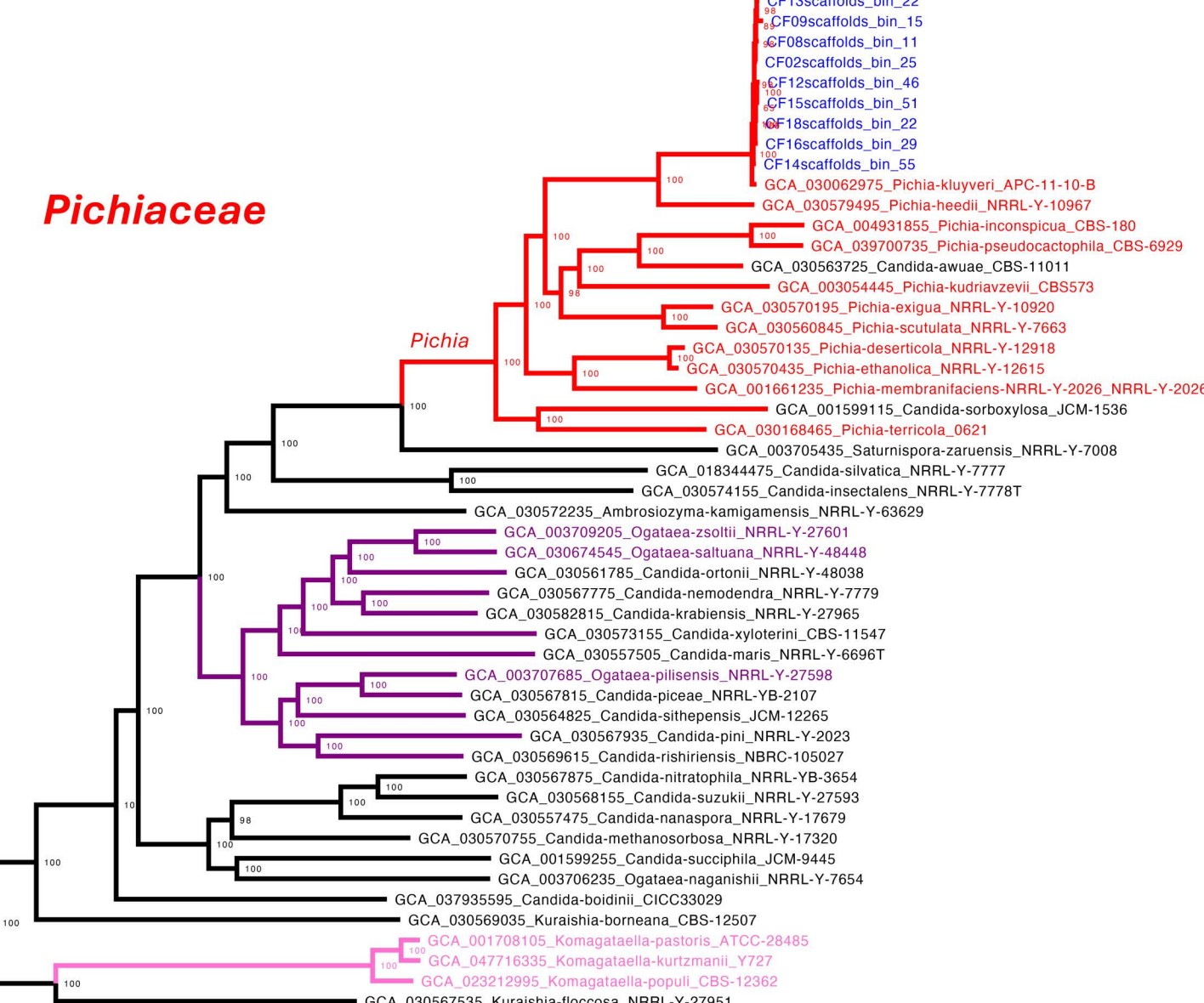

**Fig 3. Species-level phylogenomic placement of yeast MAGs within the family Pichiaceae.** The tree was inferred from a concatenated alignment of 832 conserved single-copy orthologs and includes reference genomes alongside yeast MAGs recovered from coffee fermentation metagenomes. All nodes relevant to species-level resolution are supported by ultrafast bootstrap (UFB) values of 100. Nine MAGs clustered tightly with the *Pichia kluyveri* reference genome (GCA_030062975), forming a monophyletic clade with very short intra-clade branch lengths (0.0008–0.0098) and a small genetic distance to the reference (0.0070), supporting their conspecific identity. The boundaries of the *Pichia* genus are highlighted in red, and MAG labels are shown in blue.

conspecific status. Nearby species such as *H. pseudoguilliermondii*, *H. guilliermondii*, and *H. lachancei* were clearly separated by longer branches (0.047–0.066).

The other three formed a second well-supported clade (UFB support = 100) sister to *H. meyeri*, *H. clermontiae*, *H. nectarophila*, and *H. uvarum*. Branch lengths among these MAGs (0.0044–0.0234) and to the nearest references (0.032–0.077) suggest they may represent a possibly a novel species within the genus (Fig 4).

The two MAGs assigned to Saccharomycetaceae were robustly assigned to *Torulaspora delbrueckii*, clustering with the reference genome (GCA_000243375) with UFB support = 100. Short internal branches (0.0038–0.0206) support their identity as conspecific strains. The clade was distinct from related *Torulaspora* species such as *T. franciscae*, *T. globosa*, and *T. microellipsoides* (Fig 4).

Five MAGs clustered within the order Serinales. Three of them grouped within the genus *Kurtzmaniella* (family Debary-omycetaceae), forming a strongly supported monophyletic clade (UFB support = 100) characterized by short internal branches (0.00066–0.0038 substitutions per site), indicating close genomic similarity. The closest reference genome was *Kurtzmaniella quercitrusa*, separated by a branch length of 0.0731. Based on this proximity, we assigned the MAGs to *K.*

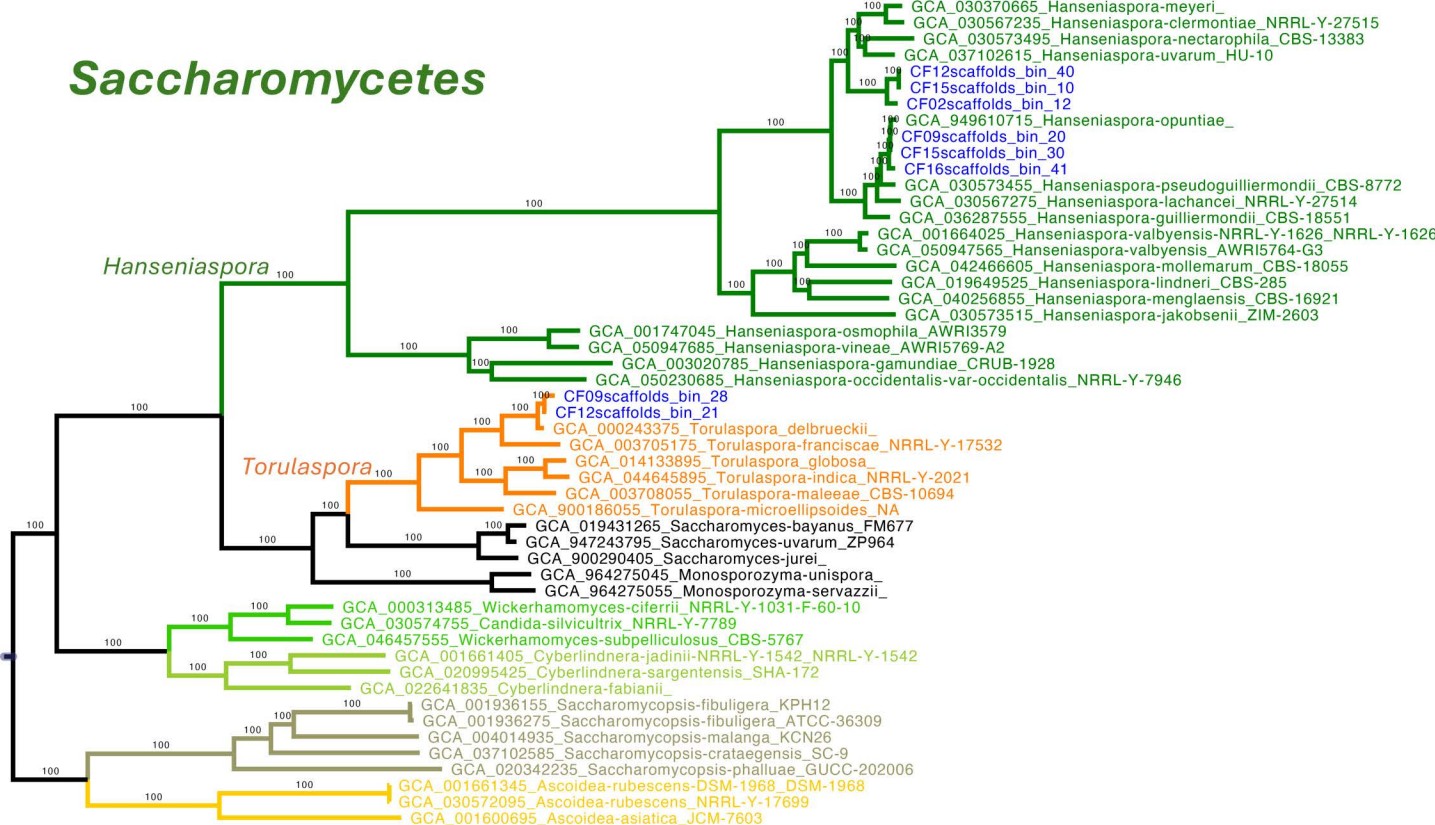

**Fig 4. Phylogenomic placement of yeast MAGs within the class Saccharomycetes, focusing on the families Saccharomycodaceae and Saccharomycetaceae.** The tree was constructed from a concatenated alignment of 832 conserved single-copy orthologs, including yeast MAGs recovered from coffee fermentation metagenomes and publicly available reference genomes. Six MAGs clustered within the genus *Hanseniaspora* (Saccharomycodaceae) in two well-supported clades (UFB support = 100). One clade grouped with H. opuntiae, with short branch lengths. The second clade, sister to *H. meyeri, H. clermontiae, H. nectarophila,* and *H. uvarum*, showed intra-clade distances of 0.0044–0.0234 and inter-clade distances of 0.032–0.077, suggesting the presence of a potentially novel species. Two additional MAGs clustered with *Torulaspora delbrueckii* (Saccharomycetaceae), forming a distinct, well-supported clade. MAGs are labeled in blue; genus boundaries are highlighted with colored lines.

*quercitrusa*. In contrast, other *Kurtzmaniella* species such as *K. natalensis*, *K. fragi*, and *K. cleridarum* were more distantly related, with branch lengths ranging from 0.112 to 0.167 (Fig 5).

The two remaining MAGs, CF02scaffolds_bin_4 and CF14scaffolds_bin_31, did not cluster within any well-characterized yeast genera. CF02scaffolds_bin_4 grouped with *Candida oleophila* (UFB support = 100) but exhibited a relatively long branch length (0.1082), suggesting it may represent a divergent strain or an uncharacterized taxon. Similarly, CF14scaffolds_bin_31 clustered with *Candida corydali* (UFB support = 100), with a branch length of 0.0386, and was part of a broader clade that included *C. sojae*. Due to their long branches and uncertain phylogenetic placement, both MAGs were conservatively assigned to Debaryomycetaceae (Fig 5).

## Yeast relative abundance and prevalence

By combining sequencing depth with phylogenomic placement, we estimated the relative abundance of MAGs using a scatterplot of scaffold length versus sequencing depth. Each MAG was color-coded according to its taxonomic assignment, while non-yeast scaffolds were shown in gray. This visualization effectively distinguished yeast MAGs from the background metagenomic sequences

*Pichia kluyveri* was the most widespread yeast, detected in all coffee fermentation samples, and was the most abundant species in every sample except CF09, where it was the least abundant. In that sample, *Kurtzmaniella* emerged as the dominant yeast. *Pichia kluyveri* MAGs consistently exhibited higher average sequencing depths compared to other identified yeasts, ranging from 5.91× to 142 ×. Particularly high abundances were observed in samples CF08 (142×), CF15 (65.7×), CF02 (55.7×), CF16 (54.6×), CF18 (45.9×), CF12 (45.5×), and CF14 (42.5×), highlighting its dominant role in the fermentation microbiota.

*Hanseniaspora* was the second most frequent genus, represented in six samples. Three MAGs assigned to *Hanseniaspora opuntiae* were recovered from CF09, CF15, and CF16, with moderate sequencing depths (10–14.9×), while other *Hanseniaspora* MAGs, likely representing distinct taxa, were found in CF02, CF12, and CF15 (9.8–41.7×). CF15 showed to distinct species of *Hanseniaspora*. *Kurtzmaniella* species were recovered from three samples (CF02, CF09, CF15), with consistent but lower depth values (8.7–16.3×), indicating lower relative abundance. *Torulaspora delbrueckii* MAGs were found in CF09 and CF12 with relatively low depths (7.2× and 8.0 ×, respectively), suggesting more limited representation. Lastly, two MAGs placed near *Candida* species—*C. oleophila* and *C. corydali*—were found in CF02 and CF14, respectively, with moderate sequencing depths (6.5× and 20.3×). Overall, these results indicate that while *Pichia kluyveri* dominates the yeast community in terms of both prevalence and abundance, other genera like *Hanseniaspora*, *Kurtzmaniella*, and *Torulaspora* contribute species- and sample-specific diversity to the coffee fermentation microbiome (Fig 6). The highest yeast MAG diversity was observed in samples CF02, CF09, and CF15, each harboring four distinct yeast species, including *Pichia kluyveri*, *Hanseniaspora* spp., and *Kurtzmaniella* spp. Notably, these three genera co-occurred consistently in all three of these diverse samples. Across all samples, the most frequent combination was the co-presence of *Pichia* and *Hanseniaspora*, found in 56% of the samples, highlighting their possible dominant and recurrent role in the coffee fermentation (S3Table).

## Functional prediction of MAGs

The dominant yeast MAGs identified across all samples: *Pichia*, *Hanseniaspora, Kurtzmaniella*, and *Torulaspora*, were functionally annotated using EggNOG-mapper and KEGG databases. Metabolic reconstruction revealed pathways involved in the degradation of mucilaginous coffee components, including reducing sugars, sucrose, starch, lipids, proteins, and organic acids. These yeasts also displayed fermentative routes contributing to flavor formation, such as alcoholic and lactic fermentation, lipid degradation, and acetification.

In all four yeasts, we detected enzymes associated with ethanol fermentation, including pyruvate decarboxylase [EC:4.1.1.1], aldehyde dehydrogenase (NAD$^+$) [EC:1.2.1.3], phosphoglycerate kinase [EC:2.7.2.3], and pyruvate kinase

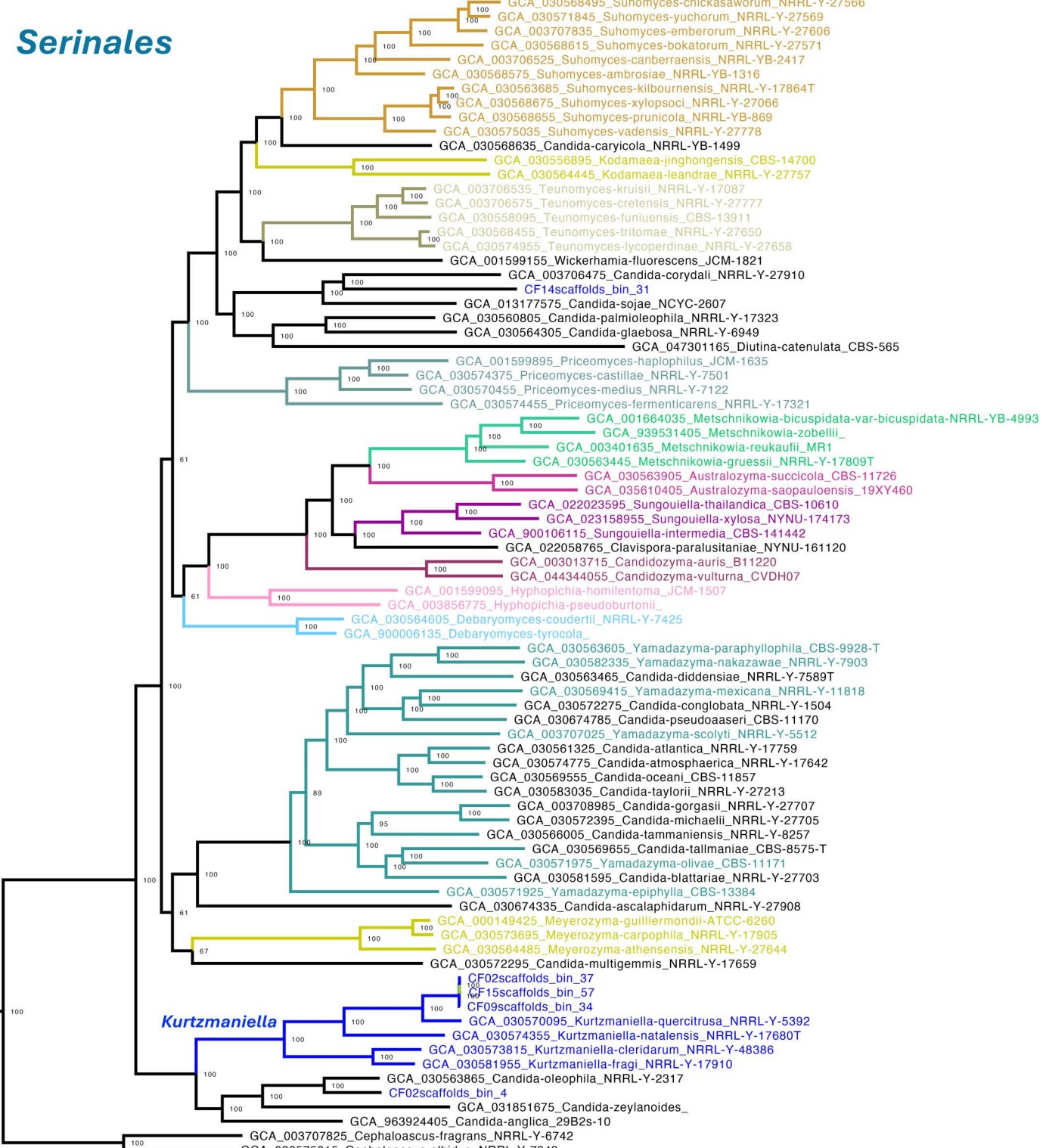

**Fig 5. Phylogenomic placement of yeast MAGs within the order Serinales, focusing on the family Debaryomycetaceae and the genus *Kurtzmaniella*.** The tree was inferred from a concatenated alignment of 832 conserved single-copy orthologs, including MAGs recovered from coffee fermentation metagenomes and relevant reference genomes. Three MAGs clustered within the genus *Kurtzmaniella*, forming a strongly supported

monophyletic clade (UFB support = 100). This clade was most closely related to the *Kurtzmaniella quercitrusa* reference genome (GCA_030570095). Two additional MAGs, CF02scaffolds_bin_4 and CF14scaffolds_bin_31, also fell within *Debaryomycetaceae* but did not group within well characterized genera. CF02scaffolds_bin_4 clustered with *Candida oleophila* (UFB support = 100) but was separated by a relatively long branch (0.1082), while CF14scaffolds_bin_31 clustered with *Candida corydali* (UFB support = 100) with a branch length of 0.0386, forming a broader clade with *Candida sojae*. Due to their genetic distance and unresolved placement, both were conservatively labeled as *Debaryomycetaceae* sp. MAGs are shown in blue; genus boundaries are marked with colored lines.

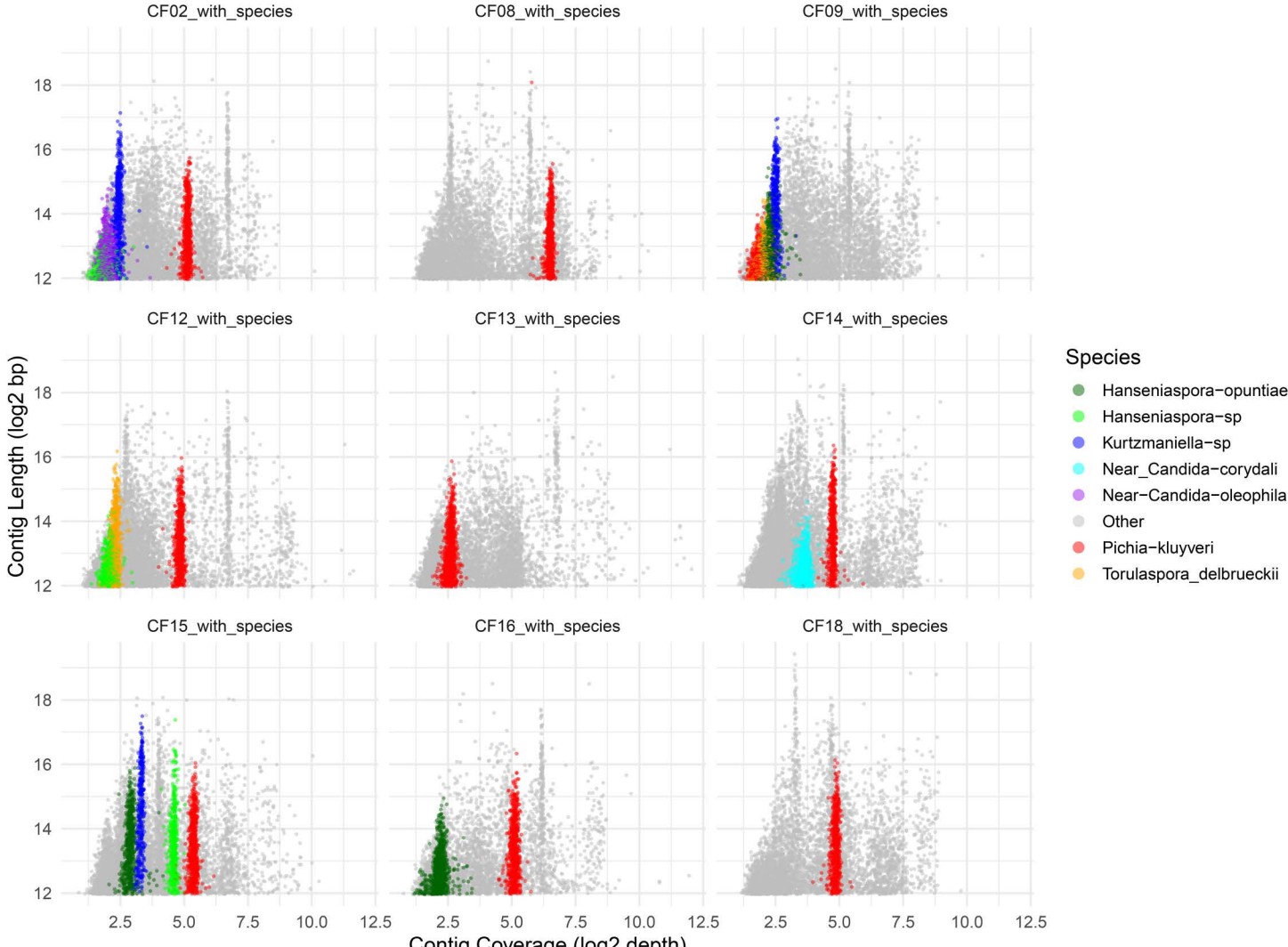

**Fig 6. Visualization of scaffold coverage and length to estimate the relative abundance of yeast MAGs across coffee fermentation samples.** Each point represents a contig and is color-coded according to its species-level taxonomic assignment based on phylogenomic analysis. Contigs not assigned to yeast MAGs are shown in gray. *Pichia kluyveri* MAGs are highlighted in red, *Hanseniaspora opuntiae* in dark green, other *Hanseniaspora* sp. in green, *Torulaspora delbrueckii* in orange, *Kurtzmaniella* in blue, Debaryomycetaceae sp. near *Candida corydali* in cyan, and Debaryomycetaceae sp. near *Candida oleophila* in purple. Only contigs ≥ 4 kb are displayed to reduce noise from short sequences. Each panel (facet) represents a different sample, enabling visual comparison of yeast MAG abundance within the broader metagenomic context.

[EC:2.7.1.40]. Carbohydrate metabolic pathways, such as fructose and mannose metabolism, galactose metabolism, and starch and sucrose metabolism, support their ability to convert simple sugars (glucose, fructose, galactose, and sucrose) into ethanol and other volatile metabolites. Genes encoding invertase and β-glucosidases indicate a strong capacity to hydrolyze complex polysaccharides and glycosides, releasing fermentable sugars and aroma precursors from coffee mucilage.

Additionally, the four yeasts exhibited genes associated with the production of citrate, aldehydes, and acetic acid. *Kurtzmaniella* also encoded enzymes linked to formic acid biosynthesis, while *Torulaspora* showed potential for lactic and malic acid production, suggesting species-specific metabolic specialization that may influence the chemical profile and sensory attributes of fermented coffee.

## Discussion

Yeasts of the subphylum Saccharomycotina are among the most challenging fungal taxa to classify using conventional microbiological methods, with widespread taxonomic inconsistencies across the group [34]. *Candida* is a prominent example, having a particularly complex and chaotic taxonomic history; the genus is polyphyletic, with species distributed across multiple families [35,36]. Although some yeast genera present fewer classification issues, polyphyletic patterns are common across several lineages within the subphylum [34].

To address the taxonomic complexity of yeasts in coffee fermentation and overcome limitations of traditional culture-based methods, we implemented a culture-independent shotgun metagenomic approach [17,37]. This strategy circumvents cultivation bias and enabled the recovery of metagenome-assembled genomes (MAGs) representing the dominant yeasts present in industrial-scale fermentation samples from various Colombian coffee producers. Then, we constructed a robust phylogenomic framework based on a comprehensive set of Saccharomycotina reference genomes, allowing accurate classification of candidate yeast MAGs at the genus or species level. This approach provided a high-resolution view of yeast biodiversity in coffee fermentations, addressing the limitations of conventional microbiological methods, which are often prone to misidentification. A common drawback of widely used marker-based strategies, such as ITS amplification, is that they typically allow identification only at the genus level, leaving considerable uncertainty about the exact species participating in fermentation. By contrast, our genome-resolved metagenomic framework not only enables confident species-level assignments but also helps bridge the gap between yeast community composition, their roles in the biochemical transformation of compounds during fermentation, and the resulting impact on coffee quality.

From all the fermented coffee metagenomes analyzed, we recovered between one to four yeast MAGs per sample. Overall, the recovered MAGs fell into two broad categories: (1) MAGs that matched previously described yeast species, and (2) MAGs whose placement in the phylogenomic tree, along with their genetic distance from known reference genomes, suggested they represent potentially novel taxa. This highlights a dual pattern in the yeast biodiversity associated with coffee fermentation: the presence of well-characterized species and the likely participation of undescribed lineages.

A clear trend observed across samples was the dominance of *Pichia kluyveri*, which emerged as the most abundant yeast in the fermentation systems studied in the Andean region of Colombia. This dominance was supported by both the relative abundance of its MAGs and the high average sequencing depth of its genomic scaffolds. The second most prevalent and abundant genus was *Hanseniaspora*, represented by MAGs closely matching *H. opuntiae*, as well as others that could not be confidently assigned to any known species. This suggests the presence of a potentially novel species within *Hanseniaspora*. While *Pichia* MAGs consistently pointed to a single species, *Hanseniaspora* showed higher taxonomic diversity, with two distinct taxa identified in sample CF15 alone.

*Kurtzmaniella* was detected as a subdominant taxon in three samples. Although the MAGs formed a well-supported monophyletic clade with *Kurtzmaniella quercitrusa* (GCA_030570095), the considerable phylogenetic distance from the reference genome deserves further analysis to confirm whether these MAGs belong to *K. quercitrusa* or instead represent

a novel species within the genus. Similarly, *Torulaspora delbrueckii* was found as a subdominant species in two samples. Its assignment was highly confident, with short genetic distances to the reference genome, supporting its classification.

The two Debaryomycetaceae MAGs represent another noteworthy finding. Although they were detected in only one sample each, their quality metrics and phylogenetic coherence support their validity. However, given their placement within clades currently subject to taxonomic uncertainty, further investigation will be necessary to resolve their classification.

The detection of both well-known and potentially novel yeast species, as well as the variation in subdominant taxa across coffee fermentation batches, underscores the influence of artisanal fermentation practices within the Colombian coffee industry. These differences in key fermentative organisms—yeasts—may plausibly contribute to the variability in the organoleptic profiles of different coffee batches [38], even though all farms in this study harvest the same *Coffea arabica* cultivar (L. Castillo). In contrast, the consistent presence of *Pichia kluyveri* across all samples suggests that, despite the artisanal nature of the processes, this yeast is able to persist across fermentation batches. However, its relative abundance does vary between samples.

When we asked the producers whether they inoculate the fermentation batches with yeast, all confirmed that the process is entirely spontaneous. This suggests that the origin of the yeasts may be the coffee fruit itself or environmental sources within the farm that inadvertently inoculate the fermentation tanks. In the case of *Pichia*, several studies have reported its presence on coffee fruits, supporting the hypothesis that this genus is part of the natural microbiota of the plant [39]. Under this scenario, *Pichia kluyveri* may be spontaneously introduced into the fermentation system and has potentially adapted to thrive in the coffee bean fermentation environment.

It is well established that various yeasts—including *Hanseniaspora*, *Torulaspora*, and *Pichia*—can colonize the surfaces of fruits such as grapes, *Cornus kousa* fruits [40], and mango [41]. These findings support the idea that fruit-associated yeasts can play a central role in shaping spontaneous fermentations, including those occurring in coffee processing [42,43].

*Pichia kluyveri* emerged as the most abundant and persistent yeast species across all coffee fermentation samples, underscoring its ecological dominance and potential functional importance in shaping the fermentation process. This resilience may be attributed to its ability to thrive under the harsh physicochemical conditions of fermentation, such as acidic pH, elevated osmotic pressure, and ethanol stress [44–47] including coffee [48–50]. Beyond its robustness, *P. kluyveri* is also known for its unique metabolic profile, particularly the production of acetate esters (e.g., ethyl acetate, isoamyl acetate) and medium-chain volatile compounds that contribute fruity and floral notes to fermented products [48,51]. Such metabolites are widely recognized for enhancing the organoleptic qualities of beverages and may similarly influence the aroma and flavor profile of fermented coffee [52]. Additionally, *P. kluyveri* has been shown to modulate the production of higher alcohols and organic acids, further impacting sensory complexity [51]. Its consistent presence and high sequencing depth across all fermentation batches suggest not only ecological fitness but also a potentially critical role in the biochemical transformation of coffee substrates during spontaneous fermentation. These metabolic features, combined with its robust growth across diverse fermentation conditions, make *P. kluyveri* a strong candidate for shaping the sensory characteristics of fermented coffee. The consistent prevalence of this species across samples suggests a reproducible influence on flavor development, possibly contributing to the consistency and distinctiveness of coffees produced through artisanal fermentations in Colombia.

The high number of alternate alleles observed in all *Pichia* and *Hanseniaspora* MAGs provides evidence for strain-level heterogeneity within the yeast populations inhabiting the coffee fermentation environment. This intra-species diversity suggests that multiple strains of the same species coexist and potentially interact within a single fermentation batch. Such co-occurrence likely reflects complex ecological dynamics, including competition, niche partitioning, and mutualistic interactions with other microbes or with the coffee fruit matrix itself, giving rise to complementary metabolic roles during fermentation.

This strain-level diversity may have important implications for the chemical complexity and sensory qualities of the final coffee product, as different yeast strains can differ in their production of volatile compounds, organic acids, and enzymes

involved in mucilage degradation and aroma formation. The observed heterogeneity may reflect a long-standing eco-logical association between these yeasts and *Coffea arabica* fruits, shaped by repeated cycles of natural inoculation, fermentation, and environmental selection. This scenario supports the notion that *Pichia* and *Hanseniaspora* are not only dominant taxa but also long-term symbionts of the coffee fermentation niche, potentially playing conserved functional roles across fermentation batches and regions. Both genera have previously been described as part of the natural microbiota of coffee fruits, reinforcing their ecological relevance in this fermentation process.

In Colombia, metataxonomic approaches based on ITS gene amplification have been applied to characterize yeast communities in fermented coffee samples from several farms in the Quindío Department, revealing *Pichia* and *Candida* as dominant genera, while *Hanseniaspora* appeared at lower relative abundance (1.8%) [53]. Other studies conducted in different Colombian regions across multiple fermentation times reported *Kazachstania humilis* as the most persistent species, followed by *Hanseniaspora vinae* and *Wickerhamomyces anomalus* [54].

In contrast, our study based on a metagenomic approach integrating metagenome-assembled genomes (MAGs) and high-resolution taxonomic profiling revealed a yeast community dominated by *Pichia, Kurtzmaniella, Hanseniaspora,* and *Torulaspora*. This composition differs from previously reported profiles and suggests the existence of regional or process-specific microbial consortia in Colombian coffee fermentations. The detection of *Kurtzmaniella* and *Torulaspora* as dominant genera highlights their potential ecological and metabolic relevance in spontaneous fermentations, under-scoring the importance of using metagenomics for a more comprehensive understanding of yeast diversity and its contri-bution to coffee flavor development.

During spontaneous coffee fermentation, yeasts are recognized as key contributors to both mucilage degradation and the biosynthesis of desirable metabolites that shape coffee flavor and aroma profiles [9,42,55]. Previous studies have demonstrated that yeasts produce essential volatile compounds such as ethyl acetate, isoamyl acetate, ethanol, and acetaldehyde, all of which are directly associated with fruity and floral sensory attributes [56,57]. In our study, we identified genes related to the production of key organic acids, including citric and acetic acids in all dominant MAGs of yeasts. Hight concentration of citric acid has been associated with the perception of acidity, fruity notes in coffee flavor [55], whereas low concentration of acetic acid contributes to fruity, wine-like, and mildly fermented aromas [58]. More-over, genes involved in lactic and malic acid biosynthesis were detected in metagenome-assembled genomes (MAGs) assigned to *Torulaspora*, suggesting their potential role in generating buttery and fruity flavor notes, respectively. Formic acid biosynthetic genes were identified in *Kurtzmaniella*, consistent with its reported contribution to fermented and slightly pungent aromas.

Traditionally, coffee-fermenting yeasts have been associated with high pectinolytic activity and thus presumed to contribute significantly to mucilage degradation [1]. However, our results did not reveal genes or complete metabolic pathways associated with pectin, cellulose, or hemicellulose degradation, suggesting that yeasts may play only a lim-ited or negligible role in mucilage hydrolysis under these fermentation conditions. This interpretation is consistent with the findings of Elhalis et al., who demonstrated that complete mucilage degradation can occur in the absence of yeasts, further supporting the view that yeast involvement in this process is minimal [9]. Together, these observations reinforce the hypothesis that bacterial members of the fermentation microbiota are primarily responsible for the breakdown of complex polysaccharides during coffee fermentation [6].

The detection of genes associated with the synthesis of citric, acetic, lactic, malic, and formic acids highlights the yeast influence in the sensory quality of Colombian coffee. These metabolites are linked to desirable organoleptic traits that characterize high-quality Colombian coffee.

## Conclusions

Coffee fermentation in the Department of Quindío from Colombia is strongly shaped by both physicochemical conditions and fermentation strategy, which collectively drive the composition of the yeast community. The phylogenomic analyses of

 

MAGs identified the specie *Pichia kluyveri* as the most dominant and ubiquitous in all samples, exhibiting high coverage across most fermentations, while genera such as *Hanseniaspora*, *Kurtzmaniella*, and *Torulaspora* contributed sample and method-specific diversity. The detection of potentially novel lineages within *Hanseniaspora* and the Debaryomycetaceae highlights the power of next-generation genomic approaches to uncover previously uncharacterized microbial diversity in traditional fermentations. Together, this study offers an integrate phylogenomic framework provided new resolution to the taxonomic relationships among coffee-associated yeasts, clarifying the placement of poorly classified taxa and revealing evolutionary connections that traditional markers failed to resolve.

## Supporting information

**S1 Fig. Summary of genome assembly quality metrics across all metagenomes.** Each boxplot represents the distribution of five key assembly statistics across genomes reconstructed from nine metagenomic samples: **(A)** Total assembly size (Mbp), **(B)** Number of scaffolds per assembly, **(C)** N50 value (bp), **(D)** GC content (%), and **(E)** N base content as a percentage of total assembly. Boxes represent the interquartile range, the horizontal line indicates the median, and whiskers extend to 1.5×the interquartile range. Assembly size is presented in megabase pairs (Mbp) using a scale transformation for readability.
(PDF)

**S2 Fig. Scatter plots showing contig length versus sequencing depth across the nine metagenome assemblies.** Each point represents a contig with length ≥4,000 bp. The X-axis corresponds to the $\log_2$-transformed average sequencing depth, while the Y-axis shows the $\log_2$-transformed contig length. Points are colored according to GC content binned into four categories: <30% (red), [30–45%] (blue), [45–60%] (gray), and >60% (dark green). Only contigs with sufficient length were retained to reduce noise from fragmented sequences. Each panel corresponds to a different sample (CF02–CF18.
(PDF)

**S3 Fig. GC content, sequencing depth, and assembly length of metagenomic bins across spontaneous coffee fermentation samples.** Each point represents a bin, plotted by its average sequencing depth ($\log_2$-transformed, x-axis) and total assembly length ($\log_2$-transformed, y-axis). Point size scales with assembly length. Points are colored according to GC content binned into four categories: <30% (red), [30–45%] (blue), [45–60%] (gray), and >60% (dark green). Faceted panels display individual samples (CF02 to CF18.
(PDF)

**S4 Fig. Maximum likelihood phylogenomic tree (UFB support=5000 replicates) of Saccharomycotina yeasts, including metagenome-assembled genomes (MAGs) recovered from coffee fermentation metagenomes.** The tree was constructed from a concatenated alignment of 832 conserved single-copy protein-coding genes and includes reference genomes representing all major Saccharomycotina lineages. MAGs are shown in blue to distinguish them from reference genomes. Ultrafast bootstrap (UFB) support values are shown for relevant nodes.
(PDF)

**S1 Table. Summary of sample conditions and associated physical parameters.**
(XLSX)

**S2 Table. Taxonomic classification and quality metrics of yeast metagenome-assembled genomes (MAGs) derived from coffee fermentation metagenomes.**
(XLSX)

**S3 Table. Accession numbers of yeast genomes included in the phylogenomic analysis.**
(XLSX)

**S1 Software Information. Detailed description of the software tools and command-line parameters used in this study.**
(TXT)

## Author contributions

**Conceptualization:** Aida Esther Peñuela-Martínez, Juan F. Alzate.

**Data curation:** Katherine Bedoya-Urrego, Aida Esther Peñuela-Martínez, Juan F. Alzate.

**Formal analysis:** Katherine Bedoya-Urrego, Aida Esther Peñuela-Martínez, Juan F. Alzate.

**Funding acquisition:** Aida Esther Peñuela-Martínez.

**Investigation:** Katherine Bedoya-Urrego, Aida Esther Peñuela-Martínez, Juan F. Alzate.

**Methodology:** Aida Esther Peñuela-Martínez, Juan F. Alzate.

**Project administration:** Aida Esther Peñuela-Martínez.

**Software:** Katherine Bedoya-Urrego, Juan F. Alzate.

**Supervision:** Aida Esther Peñuela-Martínez, Juan F. Alzate.

**Validation:** Aida Esther Peñuela-Martínez, Juan F. Alzate.

**Visualization:** Katherine Bedoya-Urrego, Juan F. Alzate.

**Writing – original draft:** Katherine Bedoya-Urrego, Aida Esther Peñuela-Martínez, Juan F. Alzate.

**Writing – review & editing:** Katherine Bedoya-Urrego, Aida Esther Peñuela-Martínez, Juan F. Alzate.

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
