## [Decision Letter · Decision Letter 0]

20 Oct 2025

Dear Dr. Alzate,

Thank you for submitting your manuscript to PLOS ONE. After careful consideration, we feel that it has merit but does not fully meet PLOS ONE’s publication criteria as it currently stands. Therefore, we invite you to submit a revised version of the manuscript that addresses the points raised during the review process.

**Both reviewers judged the study fairly conceived and scientifically relevant but the metagenomics approach to identify yeast species insufficiently robust and accurate. Sample set is also suggested to be broadened using cultivated taxa for comparison. Other issues are mostly related to the methodological clarifications or disagreements between the methodology and the obtained results. Hence, there are several major obstacles for this manuscript to be further considered for publication in this stage. I advise the authors to meticulously analyze both reviewers' reports and to thoroughly revise the manuscript, after additional experiments and analyses have been done.**

We look forward to receiving your revised manuscript.

Kind regards,

Branislav T. Šiler, Ph.D.

Academic Editor

PLOS ONE

Journal Requirements:

3. Please note that PLOS One has specific guidelines on code sharing for submissions in which author-generated code underpins the findings in the manuscript. In these cases, we expect all author-generated code to be made available without restrictions upon publication of the work. Please review our guidelines at https://journals.plos.org/plosone/s/materials-and-software-sharing#loc-sharing-code and ensure that your code is shared in a way that follows best practice and facilitates reproducibility and reuse.

“AEPM

This research was developed under the project: Experimental Development for the Competitiveness

of the Coffee Sector of the Department of Quindío, code 2017000100099, financed by

the General System of Royalties, Gobernación de Quindío in agreements signed with the National

Colombian Coffee Growers Federation (Cenicafé—Crossref Funder ID 100019597). No. 002-of-2020.”

6. Thank you for uploading your study's underlying data set. Unfortunately, the repository you have noted in your Data Availability statement does not qualify as an acceptable data repository according to PLOS's standards.

8 If the reviewer comments include a recommendation to cite specific previously published works, please review and evaluate these publications to determine whether they are relevant and should be cited. There is no requirement to cite these works unless the editor has indicated otherwise.

Reviewers' comments:

Reviewer's Responses to Questions

**Comments to the Author**

1. Is the manuscript technically sound, and do the data support the conclusions?

Reviewer #1: Yes

Reviewer #2: Partly

2. Has the statistical analysis been performed appropriately and rigorously?

Reviewer #1: Yes

Reviewer #2: No

3. Have the authors made all data underlying the findings in their manuscript fully available?

Reviewer #1: Yes

Reviewer #2: No

4. Is the manuscript presented in an intelligible fashion and written in standard English?

Reviewer #1: Yes

Reviewer #2: Yes

Reviewer #1: The paper entitled “Uncovering the Hidden Yeast Diversity in Fermented Coffee: Insights from a Shotgun Metagenomic Approach” by Katherine Bedoya-Urrego, Aida Esther Peñuela-Martínez, and Juan Fernando Alzate is an original descriptive study of fungal diversity in coffee fermentations. The paper is well-written, thoroughly analyzed, and clearly presented.

Although there is a long tradition of studies focusing on the culturable fraction of fungi and metabarcoding of molecular marker DNA (ITS, ribosomal RNA genes) associated with various coffee bean fermentations, the metagenomic approach—understood as the sequencing of all genes and species present in a sample or habitat—represents a novel perspective in this field. The article is original and interesting; however, in its current version, the identification of yeasts based solely on metagenomic information is insufficient for publication in PLOS ONE (IF 2.6, Q1–Q2). The arguments supporting this opinion are presented below. If the authors are able to enlarge the genomic and metagenomic analyses, the paper could be reconsidered.

In my opinion, the study would have been strengthened by including genomes from the cultivable fungal fraction. This information could help confirm the relative abundance of each species, improve species-level identifications, and enable the investigation of important genes related to relevant phenotypic traits during mixed coffee fermentation, among other opportunities.

I understand that there may be limitations in achieving species-level identification; however, in some cases, species belonging to certain clades are not mentioned. In addition to phylogenomic analyses, species-level identification should be further supported by ANI and AAI estimates, in silico DNA–DNA hybridization, and other complementary analyses. If the authors consider it appropriate and the results suggest the discovery of a potential new species, they could mention this possibility without committing to a formal description.

Beyond taxonomic analyses, the study does not fully exploit the large amount of data generated to perform comparative genomic analyses among the identified yeast species. Such analyses could provide insights into the role of widely distributed yeast species in coffee bean fermentation. Furthermore, no genes or metabolic pathways relevant to fermentation—such as extracellular enzymes, carbohydrate assimilation, secondary metabolites, fermentation metabolism, and other functional aspects—are highlighted. If the metagenomic work is intended to remain descriptive, it should also include bacterial and archaeal communities or microbiomes. It would also be valuable to compare the presence or absence of fungal genera detected by similar or different approaches in previous coffee fermentation studies with those identified in this work.

In Figure 4, sequences corresponding to several animals such as Dendroctonus, Caenorhabditis, Callosobruchus, and probably Drosophila are reported. This likely reflects an issue with inadequate data filtering. Please provide an explanation. For instance, Dendroctonus is a conifer bark beetle.

I consider that Figures 1, 2, 3, and 9 should be moved to the supplementary material section, as they only present quality analyses of massive sequencing.

Regarding taxonomic assignments (Fig. 4): Taxonomic assignment using MEGA alone is quite limited, as it relies on outdated databases. It is therefore recommended to use more updated and robust tools such as QIIME2 (https://library.qiime2.org/quickstart/moshpit).

Minor details: Line 114: Change −20 °C to −20°C (no space between the number and unit). Lines 722–723: Debaryomycetaceae sp. — the term “sp.” is unnecessary.

Finally, regarding omics approaches to explore the coffee fermentation microecosystem and its effects on cup quality, I believe that some important works that could strengthen the research’s international context are missing from the introduction or discussion.

Reviewer #2: Summary

Bedoya-Urrego et al. apply shotgun metagenomics, binning, and a phylogenomic framework based on a large set of single-copy proteins to profile Saccharomycotina yeasts in spontaneous Colombian coffee fermentations. The abstract reports 24 yeast MAGs (later 22 after filtering), dominated by Pichia kluyveri, with additional Hanseniaspora, Torulaspora, and Kurtzmaniella lineages and putative novel taxa. The study is timely and relevant; however, key reporting gaps (eukaryotic MAG quality control), internal inconsistencies (counts; assembly strategy), and methodological ambiguities prevent full evaluation.

Major revisions

1. Eukaryotic MAG quality control not reported

The manuscript does not present per-MAG completeness/duplication (e.g., BUSCO) or contamination estimates (e.g., EukCC), nor a MAG-level QC table. This is core to genome-resolved metagenomics.

Please report for each MAG: BUSCO lineage/version with C/F/M/D %, an explicit contamination estimate (e.g., EukCC), size, N50, scaffold count, GC%, median depth, and breadth.

2. Workflow clarity: per-sample assemblies vs. co-assembly

Results describe one assembly per sample; Methods describe mapping to co-assembled scaffolds. These are presented as a single workflow though they are distinct.

Please state unambiguously which assembly underlies binning, coverage, and figures (per-sample, co-assembly, or both) and align text/metrics accordingly.

3. Counts and tallies disagree

Abstract: 24 MAGs; species-level tallies sum to 22; one sentence mentions removal of a single MAG (would total 23).

Please reconcile counts across Abstract, Results, legends, and supplements.

4. Read handling for MEGAN is misdescribed

Reads are said to be “merged into longer single-end reads using “seqtk”,” which does not generate longer consensus reads. The manuscript also alternates between MEGAN on reads vs. on scaffolds.

Please correct the tool description and specify whether MEGAN used reads or scaffolds, which classifier/database (e.g., DIAMOND + NR; version/date), and LCA parameters.

5. Abundance is visualized but not quantified

A length-vs-depth scatterplot is presented as “relative abundance,” but no explicit metric or presence threshold is provided.

Please provide a per-sample, per-MAG abundance table (e.g., reads mapped per kb; TPM/RPKM) and breadth thresholds for presence.

6. Host (Coffea) depletion undocumented

Plant sequences are acknowledged; no reference genome, mapping criteria, or % removed are reported.

Please specify the Coffea reference(s), filtering criteria, and per-sample host removal summary.

7. Phylogenomic model scope vs. claims

Deep-rank statements (e.g., family non-monophyly) are paired with uniformly high UFB support under a single model description.

Please either provide a brief robustness check (partitioning and/or site-heterogeneous model) or temper claims; indicate whether key placements are model-stable.

Methodological clarifications

BLASTN thresholds. “E-value = 0” and “bitscore > 10,000” are not biologically interpretable cutoffs. Define alignment length, % identity, and coverage criteria; indicate whether reciprocal hits or AAI/ortholog identity supported species calls.

Rank terminology. The text refers to “class-level” structure while listing “-ales” groups (orders). Harmonize rank labels with current taxonomy.

Software versions/commands. Resolve the Cutadapt version mismatch (“version 3” vs “v2.10” in parameters). Provide exact versions and key command lines (trimming, assembly, binning, mapping, phylogeny) in Supplementary Methods.

Other

Flavor/functional claims. Several passages imply causal impacts on sensory outcomes; no metabolomics/sensory data are presented. Either add minimal genomic context (e.g., presence/absence of ester/pectinase loci per MAG) or consistently frame these as literature-based hypotheses.

“Megan sequencing depth” → mean sequencing depth.

“CF15 showed two distinct species of Hanseniaspora.”

Consistent capitalization (MEGAN), consistent “phylogenomics” wording, report “UFBoot support = 100%”.

Avoid mixing CFxx and Fxx for the same batches; provide a simple crosswalk table.

**Do you want your identity to be public for this peer review?** For information about this choice, including consent withdrawal, please see our Privacy Policy

Reviewer #1: **Yes:** César Hernández-Rodríguez

Reviewer #2: No

---

## [Author Response · Author response to Decision Letter 1]

2 Dec 2025

PONE-D-25-47268

Uncovering the Hidden Yeast Diversity in Fermented Coffee: Insights from a Shotgun Metagenomic Approach

PLOS ONE

Rebuttal Letter

R:/: We sincerely thank the editor and reviewers for their time and effort in carefully evaluating our manuscript, and for their thoughtful comments, constructive critiques, and valuable recommendations. We are confident that these suggestions have substantially improved the quality and clarity of the manuscript. We have made every possible effort to address all comments and enhance the manuscript accordingly.

R:/ We carefully reviewed the journal’s guidelines and revised the manuscript format accordingly.

R:/ A detailed paragraph describing the authorization for sample collection has been added to the Methods section. This is the paragraph: “Sample collection was conducted under the valid permit issued by Resolution No. 1508 of September 6, 2018. The permit was granted by the NATIONAL ENVIRONMENTAL LICENSING AUTHORITY OF COLOMBIA (ANLA) to the Post-Harvest Coffee Research Group (COL0006634) in the implementation of the project POS101031, “Characterization of the microbial composition in the fermentation of coffee produced in the department of Quindío”; This was within the broader framework of the project “Experimental Development for the Competitiveness of the Coffee Sector of the Department of Quindío”; code 2017000100099, in agreements signed with the National Colombian Coffee Growers Federation (Cenicafé—Crossref Funder ID 100019597), No. 002-of-2020”.

3. Please note that PLOS One has specific guidelines on code sharing for submissions in which author-generated code underpins the findings in the manuscript. In these cases, we expect all author-generated code to be made available without restrictions upon publication of the work. Please review our guidelines at https://journals.plos.org/plosone/s/materials-and-software-sharing#loc-sharing-code and ensure that your code is shared in a way that follows best practice and facilitates reproducibility and reuse.

R:/ No proprietary or custom scripts were generated in this study. The analyses were conducted using publicly accessible tools, including IQ-TREE, MEGAN, BLAST, MetaBAT, BUSCO, and others. The commands used were described in supplementary material (S4 Software).

4. We note that the grant information you provided in the ‘Funding Information’ and ‘Financial Disclosure’ sections do not match. When you resubmit, please ensure that you provide the correct grant numbers for the awards you received for your study in the ‘Funding Information’ section.

R:/ The funding information was reviewed and updated to align with the Financial Disclosure section.

“AEPM This research was developed under the project: Experimental Development for the Competitiveness of the Coffee Sector of the Department of Quindío, code 2017000100099, financed by the General System of Royalties, Gobernación de Quindío in agreements signed with the National Colombian Coffee Growers Federation (Cenicafé—Crossref Funder ID 100019597). No. 002-of-2020.” Please state what role the funders took in the study. If the funders had no role, please state: "The funders had no role in study design, data collection and analysis, decision to publish, or preparation of the manuscript." If this statement is not correct you must amend it as needed.

R:/ The text has been revised accordingly, and all funding-related information has been removed from the manuscript. In addition, the suggested sentence has been included in the submission system as a comment addressed to the editor.

6. Thank you for uploading your study's underlying data set. Unfortunately, the repository you have noted in your Data Availability statement does not qualify as an acceptable data repository according to PLOS's standards. At this time, please upload the minimal data set necessary to replicate your study's findings to a stable, public repository (such as figshare or Dryad) and provide us with the relevant URLs, DOIs, or accession numbers that may be used to access these data. For a list of recommended repositories and additional information on PLOS standards for data deposition, please see https://journals.plos.org/plosone/s/recommended-repositories.

R:/ The Data Availability information included in the Methods section has been revised to comply with the journal’s guidelines. The sentence “The raw data can be accessed through the following link: https://www.ncbi.nlm.nih.gov/sra/PRJNA1305218. The sample identifiers are SRR34971509, SRR34971508, SRR34971516, SRR34971515, SRR34971514, SRR34971513, SRR34971512, SRR34971511, and SRR34971510” was added.

R:/ Captions for supplementary Figure 1 and supplementary tables were added to the manuscript. This caption was added in the end of the manuscript.

“S1 Table. Summary of sample conditions and associated physical parameters.

S2 Table. Accession numbers of yeast genomes included in the phylogenomic analysis.

S3 Table. Taxonomic classification and quality metrics of yeast metagenome-assembled genomes (MAGs) derived from coffee fermentation metagenomes.”.

Comments to the Author

5. Review Comments to the Author

Reviewer #1: The paper entitled “Uncovering the Hidden Yeast Diversity in Fermented Coffee: Insights from a Shotgun Metagenomic Approach” by Katherine Bedoya-Urrego, Aida Esther Peñuela-Martínez, and Juan Fernando Alzate is an original descriptive study of fungal diversity in coffee fermentations. The paper is well-written, thoroughly analyzed, and clearly presented.

Although there is a long tradition of studies focusing on the culturable fraction of fungi and metabarcoding of molecular marker DNA (ITS, ribosomal RNA genes) associated with various coffee bean fermentations, the metagenomic approach—understood as the sequencing of all genes and species present in a sample or habitat—represents a novel perspective in this field. The article is original and interesting; however, in its current version, the identification of yeasts based solely on metagenomic information is insufficient for publication in PLOS ONE (IF 2.6, Q1–Q2). The arguments supporting this opinion are presented below. If the authors are able to enlarge the genomic and metagenomic analyses, the paper could be reconsidered. In my opinion, the study would have been strengthened by including genomes from the cultivable fungal fraction. This information could help confirm the relative abundance of each species, improve species-level identifications, and enable the investigation of important genes related to relevant phenotypic traits during mixed coffee fermentation, among other opportunities.

I understand that there may be limitations in achieving species-level identification; however, in some cases, species belonging to certain clades are not mentioned. In addition to phylogenomic analyses, species-level identification should be further supported by ANI and AAI estimates, in silico DNA–DNA hybridization, and other complementary analyses. If the authors consider it appropriate and the results suggest the discovery of a potential new species, they could mention this possibility without committing to a formal description.

R:/ We thank the reviewer for this valuable comment and fully agree that integrating genomes from cultivated isolates could further enrich our understanding of the yeast community in coffee fermentation. However, we would like to emphasize that MAGs provide unique and complementary insights that are not attainable through culture-based approaches alone. First, MAGs capture the genomic diversity of yeast taxa that often escape isolation using traditional methods. Many yeasts involved in spontaneous fermentations, especially those from the early stages or transient populations, are difficult to recover in pure culture due to fastidious growth requirements, competition, or dependence on specific microbial interactions. Thus, the metagenomic framework allows the recovery and genomic characterization of previously unrecognized or novel lineages, as evidenced in our study by MAGs that did not match any currently described species. Second, MAGs enable a genome-resolved view of the active community structure, providing more accurate estimates of relative abundance within the fermentation matrix. This avoids cultivation bias and reflects the actual in-situ representation of each yeast species in the process.

While cultivation-based approaches remain invaluable for experimental validation, metagenomics offers an ecologically relevant, culture-independent perspective that broadens our understanding of microbial participation in complex fermentations like coffee. We view our MAG-based analysis as a complementary and foundational framework upon which future studies integrating cultivable isolates and phenotypic analyses can build.

We fully agree with the reviewer on the importance of confirming taxonomic assignments using the most current and scientifically supported approaches. For this reason, we selected a phylogenomic strategy, which is considered the gold standard for taxonomic classification, as it provides a robust framework for accurate evolutionary inference based on orthologous loci across genomes and topological context. ANI is other well supported strategy for taxonomic assignment; however, ANI thresholds and tools are primarily optimized for prokaryotic genomes, and their applicability to yeast species remains limited and not yet standardized. For these reasons, we prioritized a phylogenomic approach.

Beyond taxonomic analyses, the study does not fully exploit the large amount of data generated to perform comparative genomic analyses among the identified yeast species. Such analyses could provide insights into the role of widely distributed yeast species in coffee bean fermentation. Furthermore, no genes or metabolic pathways relevant to fermentation—such as extracellular enzymes, carbohydrate assimilation, secondary metabolites, fermentation metabolism, and other functional aspects—are highlighted.

R:/ Metabolic analyses were incorporated into the Results and Discussion sections. Comparative analyses of the metabolic potential among the yeast MAGs were conducted to identify functional and genomic distinctions between the detected species.

If the metagenomic work is intended to remain descriptive, it should also include bacterial and archaeal communities or microbiomes. It would also be valuable to compare the presence or absence of fungal genera detected by similar or different approaches in previous coffee fermentation studies with those identified in this work.

R:/ We thank the reviewer for this valuable comment. We fully agree that including bacterial and archaeal communities would provide a more comprehensive view of the coffee fermentation microbiome. However, our study was specifically designed to address a current knowledge gap regarding the yeast profile in coffee fermentation, using a metagenomic and high-resolution taxonomic approach. The bacterial component represents a highly complex system that extends beyond the scope of this work and would require a dedicated and independent analysis. We believe that focusing on the yeast fraction allows us to provide novel and detailed insights into their metabolic potential and ecological role during coffee fermentation.

We appreciate the reviewer’s insightful suggestion of compare our results with other studies in Colombia. In response, we have included in the Discussion a comparison with local studies that evaluated yeast profiles in fermented coffee samples using a metataxonomic approach. This addition allows us to contextualize our metagenomic findings with previously reported fungal community compositions obtained through different methodologies.

In Figure 4, sequences corresponding to several animals such as Dendroctonus, Caenorhabditis, Callosobruchus, and probably Drosophila are reported. This likely reflects an issue with inadequate data filtering. Please provide an explanation. For instance, Dendroctonus is a conifer bark beetle.

R:/ We thank the reviewer for this valuable observation, as we agree that it is important to carefully consider potential false positives in metagenomic-based studies. We would like to clarify that this work is based on shotgun metagenomic sequencing, in which DNA from all organisms present in the sample—including non-target eukaryotes—can be detected. Given that the coffee fermentation on farms is performed under artisanal and open conditions, it is expected that reads from environmental organisms such as nematodes and arthropods may appear in the metagenomic dataset.

Therefore, the MEGAN assignments should not be interpreted as definitive taxonomic identifications but rather as a broad overview of the taxonomic composition of the sample. The detection of taxa such as Drosophila, Caenorhabditis, or Dendroctonus does not imply their active participation in the fermentation process but instead could reflects the presence of environmental or incidental invertebrate DNA. Our intention in presenting the MEGAN results was to provide a comprehensive view of the total DNA diversity within the fermentation environment.

However, since the focus of this study is on yeasts, we validated and refined their taxonomic assignments using phylogenomics, which currently represents the gold standard for high-resolution and accurate classification of microbial genomes, including MAGs.

I consider that Figures 1, 2, 3, and 9 should be moved to the supplementary material section, as they only present quality analyses of massive sequencing.

R:/ Figures 1,2,3 were moved to supplementary material according to the reviewer suggestion. However, we considered that the Figure 9 is relevant for the results since it represents the taxonomy and relative abundance of MAGs that is an important axis of this work. For this reason, we would like to retain the figure within the Results section

Regarding taxonomic assignments (Fig. 4): Taxonomic assignment using MEGA alone is quite limited, as it relies on outdated databases. It is therefore recommended to use more updated and robust tools such as QIIME2 (https://library.qiime2.org/quickstart/moshpit).

R:/ We agree with this assessment; accurate taxonomic assignment is essential for this type of analysis. The alignment-based taxonomic methods (such as MEGAN, QIIME, or other similarity-based tools) are inherently limited by database completeness and sequence homology thresholds, which can result in both false-positive and false-negative identifications. For this reason, these methods were used only as a preliminary exploratory step, whereas the final classification of yeast MAGs was established through robust phylogenomic reconstruction.

Minor details: Line 114: Change −20 °C to −20°C (no space between the number and unit). Lines 722–723: Debaryomycetaceae sp. — the term “sp.” is unnecessary.

R:/ Corrections were made in the manuscript.

Finally, regarding omics approaches to explore the coffee fermentation microecosystem and its effects on cup quality, I believe that some important works that could strengthen the research’s internatio

---

## [Decision Letter · Decision Letter 1]

11 Jan 2026

Dear Dr. Alzate,

Thank you for submitting your manuscript to PLOS ONE. After careful consideration, we feel that it has merit but does not fully meet PLOS ONE’s publication criteria as it currently stands. Therefore, we invite you to submit a revised version of the manuscript that addresses the points raised during the review process.

**Several minor, mostly technical omissions still remained in the revised manuscript. Please address them as suggested by Reviewer #2.**

We look forward to receiving your revised manuscript.

Kind regards,

Branislav T. Šiler, Ph.D.

Academic Editor

PLOS One

Journal Requirements:

Reviewers' comments:

Reviewer's Responses to Questions

**Comments to the Author**

Reviewer #2: (No Response)

2. Is the manuscript technically sound, and do the data support the conclusions?

Reviewer #2: Yes

3. Has the statistical analysis been performed appropriately and rigorously?

Reviewer #2: Yes

4. Have the authors made all data underlying the findings in their manuscript fully available?

Reviewer #2: Yes

5. Is the manuscript presented in an intelligible fashion and written in standard English?

Reviewer #2: Yes

Reviewer #2: Thank you for the thorough revision. The manuscript is substantially improved and most major concerns have been addressed (workflow clarity, MAG counts, MEGAN read handling, and inclusion of per-MAG BUSCO and TPM outputs in the Supplementary material). The study is timely and the phylogenomic framework is a strength.

A small number of points still need tightening before acceptance: (i) please add an explicit MAG contamination estimate (e.g., EukCC or comparable) and consider reporting read-mapping breadth to each MAG (breadth can be calculated against the MAG scaffolds themselves, not only against external references); (ii) resolve remaining reproducibility inconsistencies (notably the Cutadapt version mismatch) and ensure database/version/date details are complete; (iii) revise the BLAST screening description to include interpretable criteria (alignment length, % identity, and coverage), even if phylogenomics is the primary basis for species assignment; and (iv) harmonize taxonomic rank terminology and ensure any residual “non-monophyly” language is tempered unless supported by robustness checks. With these minor clarifications, I would support publication.

**Do you want your identity to be public for this peer review?** For information about this choice, including consent withdrawal, please see our Privacy Policy

Reviewer #2: No

---

## [Author Response · Author response to Decision Letter 2]

21 Jan 2026

PONE-D-25-47268R1

Uncovering the Hidden Yeast Diversity in Fermented Coffee: Insights from a Shotgun Metagenomic Approach

PLOS One

Rebuttal Letter

We sincerely thank the editor and reviewers for their constructive comments and valuable suggestions, which have significantly improved the quality and clarity of the manuscript. We have addressed all comments accordingly.

(i) Please add an explicit MAG contamination estimate (e.g., EukCC or comparable) and consider reporting read-mapping breadth to each MAG (breadth can be calculated against the MAG scaffolds themselves, not only against external references)

R/: Thanks for the recommendation. We perform the analysis with the EukCC and report the contamination and completeness in the S3 table (column R and Q). The breath of coverage (%) of each MAG also was reported (column P).

(ii) Resolve remaining reproducibility inconsistencies (notably the Cutadapt version mismatch) and ensure database/version/date details are complete.

R/: Cutadapt version was revised, and the correction was made in Material and Methods section

(iii) Revise the BLAST screening description to include interpretable criteria (alignment length, % identity, and coverage), even if phylogenomics is the primary basis for species assignment

R/: The BLAST filtering parameters (E-value = 0 and bit score ≥ 10,000) resulted in alignment blocks of at least 2,000 bp. We verified that the minimum sequence identity of these aligned blocks relative to their respective reference genomes was 70%. For the phylogenomic analysis, single-copy protein-coding genes were predicted using AUGUSTUS, and orthology inference was performed with SonicParanoid, as described in the Methods section. This annotation and orthology framework provides robust support for the phylogenomic analyses presented in this study.

(iv) (iv) harmonize taxonomic rank terminology and ensure any residual “non-monophyly” language is tempered unless supported by robustness checks. With these minor clarifications, I would support publication.

(v) R/: The sentence has been removed from the results section.

---

## [Editor Report · Decision Letter 2]

22 Jan 2026

Uncovering the Hidden Yeast Diversity in Fermented Coffee: Insights from a Shotgun Metagenomic Approach

PONE-D-25-47268R2

Dear Dr. Alzate,

We’re pleased to inform you that your manuscript has been judged scientifically suitable for publication and will be formally accepted for publication once it meets all outstanding technical requirements.

Kind regards,

Branislav T. Šiler, Ph.D.

Academic Editor

PLOS One
---

## [Editor Report · Acceptance letter]

PONE-D-25-47268R2

PLOS One

Dear Dr. Alzate,

I'm pleased to inform you that your manuscript has been deemed suitable for publication in PLOS One. Congratulations! Your manuscript is now being handed over to our production team.

Kind regards,

on behalf of

Dr. Branislav T. Šiler

Academic Editor

PLOS One